

# Deep-sea water displacement from a turbidity current induced by the Super Typhoon Hagibis

Shinsuke Kawagucci[1,2], Tetsuya Miwa[2,3], Dhugal J. Lindsay[4,5], Eri Ogura[4,5], Hiroyuki Yamamoto[2,6], Kenichiro Nishibayashi[7], Hiroyuki Yokooka[7], Shotaro Nishi[7], Ayu Takahashi[8] and Sangkyun Lee[8]

[1] Super-cutting-edge Grand and Advanced Research (SUGAR) Program, Institute for Extra-cutting-edge Science and Technology Avant-garde Research (X-star), Japan Agency for Marine-Earth Science and Technology (JAMSTEC), Yokosuka, Japan

[2] Project Team for Developing Innovative Technologies for Exploration of Deep-Sea Resources, Japan Agency for Marine-Earth Science and Technology (JAMSTEC), Tokyo, Japan

[3] Institute for Marine-Earth Exploration and Engineering (MarE3), Japan Agency for Marine-Earth Science and Technology (JAMSTEC), Yokosuka, Japan

[4] Institute for Extra-cutting-edge Science and Technology Avant-garde Research (X-star), Advanced Science-Technology Research (ASTER) Program, Japan Agency for Marine-Earth Science and Technology (JAMSTEC), Yokosuka, Japan

[5] Graduate School of Nanobioscience, Yokohama City University, Yokohama, Japan

[6] Research Institute for Global Change (RIGC), Marine Biodiversity and Environmental Assessment Research Center (BioEnv), Japan Agency for Marine-Earth Science and Technology (JAMSTEC), Yokosuka, Japan

[7] Research and Development Partnership for Next Generation Technology of Marine Resources Survey (J-MARES)/ IDEA Consultants, Tokyo, Japan

[8] Research and Development Partnership for Next Generation Technology of Marine Resources Survey (J-MARES) / JGI, Inc., Tokyo, Japan

Corresponding author
Shinsuke Kawagucci,
kawagucci@jamstec.go.jp

## ABSTRACT

Turbidity currents are the main drivers behind the transportation of terrestrial sediments to the deep sea, and turbidite deposits from such currents have been widely used in geological studies. Nevertheless, the contribution of turbidity currents to vertical displacement of seawater has rarely been discussed. This is partly because until recently, deep-sea turbidity currents have rarely been observed due to their unpredictable nature, being usually triggered by meteorological or geological events such as typhoons and earthquakes. Here, we report a direct observation of a deep-sea turbidity current using the recently developed Edokko Mark 1 monitoring system deployed in 2019 at a depth of 1,370 m in Suruga Bay, central Japan. A turbidity current occurred two days after its probable cause, the Super Typhoon Hagibis (2019), passed through Suruga Bay causing devastating damage. Over a period of 40 hours, we observed increased turbidity with turbulent conditions confirmed by a video camera. The turbidity exhibited two sharp peaks around 3:00 and 11:00 on October 14 (Japan Standard Time). The temperature and salinity characteristics during these high turbidity events agreed with independent measurements for shallow water layers in Suruga Bay at the same time, strongly suggesting that the turbidity current caused vertical displacement in the bay's water column by transporting warmer and shallower waters downslope of the canyon. Our results add to the previous few examples that show meteorological and geological events may have significant contributions in the transportation of shallower seawater to the deep sea. Recent technological developments pertaining to the Edokko Mark 1 and

similar devices enable straightforward, long-term monitoring of the deep-seafloor and will contribute to the understanding of similar spontaneous events in the deep ocean.

# INTRODUCTION

Turbidity currents are particle-laden, gravity-driven flows that efficiently transport terrigenous sediment and organic matter, as well as benthic fauna and microplastics, to the deep sea (*Meiburg & Kneller, 2010*; *Sen et al., 2017*; *Pohl et al., 2020*). The damaging of seafloor fiber-optic cables by turbidity currents causes serious disruption of social activities (*Pope et al., 2017*). The sedimentary deposits resulting from turbidity currents are known as turbidites and have been widely used in geological studies (e.g., *Piper & Normark, 2009*). For example, dating of the repeated turbidite sequences in the accretionary prism provide decisive evidence for tectonic evolution at the convergent plate boundary (*Taira, 2001*). Despite these multidimensional interests, direct observation of turbidity currents in the deep sea had been rare until the beginning of this century (*Talling, Paull & Piper, 2013*) due to their unpredictable nature and the limited accessibility resulting from challenges and ship time costs of deep-sea expeditions although numbers of the observation have increased during the last decade (*Xu, 2010*; *Liu et al., 2012*; *Khripounoff et al., 2012*; *Hughes Clarke, 2016*; *Paull et al., 2018*; *Wang et al., 2020*; *Heerema et al., 2020*; *Hage et al., 2019*; *Normandeau et al., 2019*; *Lintern, Hill & Stacey, 2016*; *Simmons et al., 2020*).

Limited in situ direct observations provide the only clues to understand the possible impacts of deep-sea turbidity currents. To date, oceanographic CTD sensors deployed on seafloor observatories have revealed seawater temperature increases associated with deep-sea turbidity currents (*Khripounoff et al., 2012*; *Wang et al., 2020*). A deep-sea mooring observatory off Taiwan has repeatedly captured typhoon-triggered turbidity currents and subsequent temperature increases over a period of 3.5 years (*Zhang et al., 2018*). Cabled observatories placed on the deep seafloor have also captured turbidity currents triggered by earthquakes, such as at the Kuril subduction zone (*Mikada et al., 2006*) and off Hatsushima Island in central Japan (*Kasaya et al., 2009*). These observations were coupled with temperature increases, suggesting low-density, shallow seawater were transported into deep depths as the interstitial water of turbidity currents, against the density stratification of the water column (*Kao et al., 2010*). However, whether or not turbidity currents routinely cause such vertical displacement of seawater remains unclear due to the limited number of observations.

In addition to earthquakes, powerful tropical cyclones can also provide opportunities for the direct observation of deep-sea turbidity currents (e.g., *Liu et al., 2012*; *Pope et al., 2017*; *Sequeiros et al., 2019*). On October 12 in 2019 (all timestamps in this paper are presented in JST: Japan Standard Time, UTC+9:00), the extremely large Super Typhoon Hagibis (2019) struck the main island of Japan, leading to a total of 86 deaths (*Fire*

and Disaster Management Agency of Japan, 2019) and the cancellation of three Rugby World Cup matches. Hagibis reached its peak intensity with a minimum atmospheric pressure of 915 hPa over the Philippine Sea on October 7, and passed through Suruga Bay with an atmospheric pressure of 955 hPa at 18:00 on October 12 (Fig. 1A) (*Japan Meteorological Agency, 2019a*). During the passing of Hagibis, a maximum sea-level departure of 224 cm and a 24-hour cumulative precipitation of up to 760 mm per m² were recorded on the Izu Peninsula on the east coast of Suruga Bay, and a warning for severe flooding was issued for the riverine area until the morning of October 13 (*Shizuoka Local Meteological Office, 2019*; *Takemi & Unuma, 2020*). A global seafloor cable-break database analysis demonstrated that tropical cyclones can trigger deep-sea turbidity currents several days after the cyclone's passing (*Pope et al., 2017*). We thus attempted a direct observation of the Hagibis-linked deep turbidity current in Suruga Bay two days after the passing of Hagibis. Here, we report visual and oceanographic properties of this in situ turbidity current, captured by the free-fall-type deep-sea observatory system Edokko Mark 1 (Fig. 2).

## MATERIALS & METHODS

The Edokko Mark 1 (Type HSG) is an all-in-one, free-fall, stand-alone deep-sea monitoring system (*Miwa et al., 2016*; *Japan Agency for Marine-Earth Science and Technology, 2017*), provided by Okamoto Glass Co. Ltd. (https://ogc-jp.com/en/). The Edokko Mark 1 and other similar systems (*Gallo et al., 2020*; *Clare et al., 2020*) increase our capability for the observation of deep-sea turbidity currents when typhoons, earthquakes, or tsunamis impact marginal seas.

The Edokko Mark 1 has a main frame with three glass spheres containing the main computer, HD video cameras, 2,400–4,000 lumen LED lights, transponder, and batteries. The main frame is further equipped with an acoustic releaser for the ballast weight at the base, a floating glass sphere with a radio beacon and flasher at the top, and a 75 cm long, 50 cm wide PVC arm for image-based measurements in the front (Fig. 2). Edokko Mark 1 is capable of monitoring for up to three months, owing to reduced battery consumption from a programmable long-term monitoring mode allowing intermittent, periodic observations. In this study, Edokko Mark 1's electric system was continuously powered until 15:23 on 14 October, after which it entered a long-term monitoring mode with observations carried out for 1 minute every 30 minutes until shipboard recovery at 10:20 on 16 October. In the set up used in the present study, a CTD profiler (RINKO profiler, JFE Advantech) and a turbidity meter (ASTD2XTU, JFE Advantech Co., Ltd.) were further attached to the main frame at 1.2 m above the seafloor (Fig. 2). The unit of turbidity, FTU (formazine turbidity unit), is defined as 1 FTU = turbidity where 1 mg of formazin is homogeneously suspended in 1 L of water.

Edokko Mark 1 landed on the seafloor at a depth of 1,370 m at the mouth of Heda Canyon, in the northeastern region of Suruga Bay (35°59.63′N –138°40.30′E). The topography of Heda Canyon is characterized by a narrow and winding valley (Fig. 1B) that suggests the repeated occurrence of deep-sea turbidity currents in the past (e.g.,
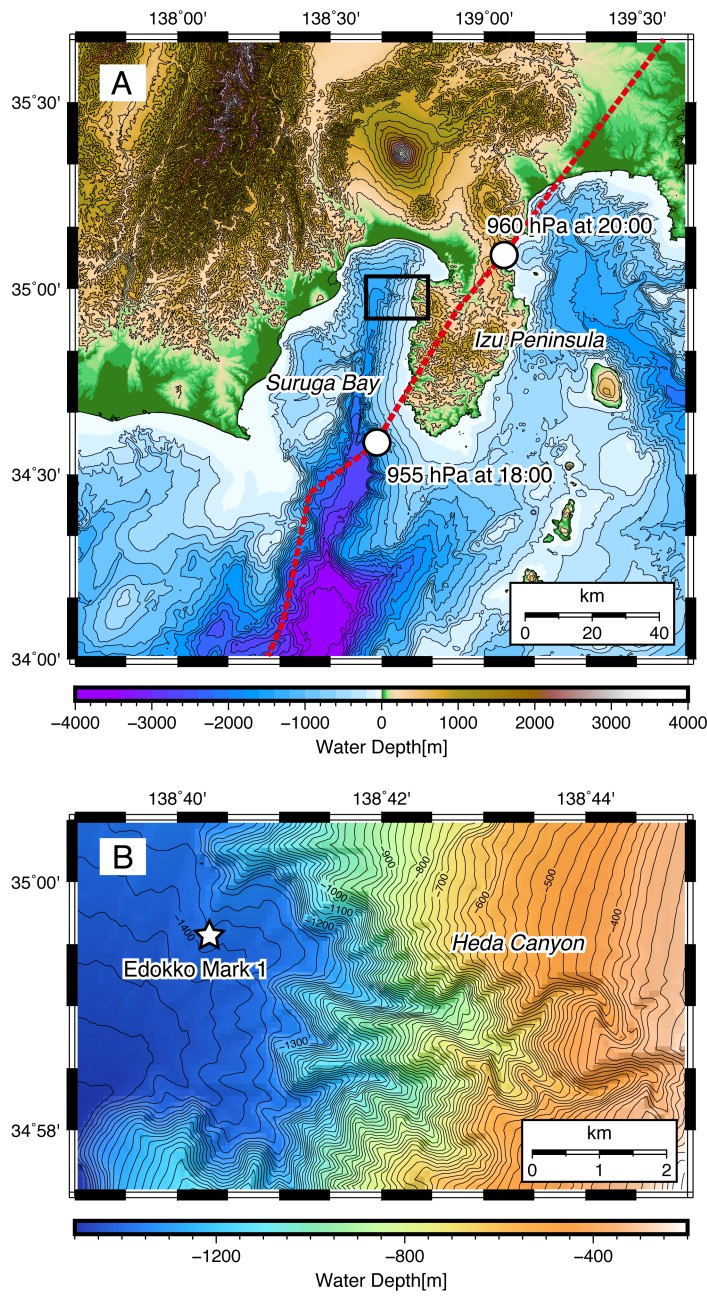

**Figure 1** **Topography around (A) Suruga Bay and (B) the Heda Canyon.** Red broken line shows a track of Typhoon Hagibis on 12 October 2019.

*Azpiroz-Zabala et al., 2017*), probably associated with floods from the Heda River located on the Izu Peninsula (Fig. 1A).

Still images were extracted from the video recordings at a 1 Hz frequency using OpenCV (cv2) within a custom-built Python script (https://github.com/dhugallindsay/Image-based-turbidity-flow-detection). Each 1 minute-long recording event contained several

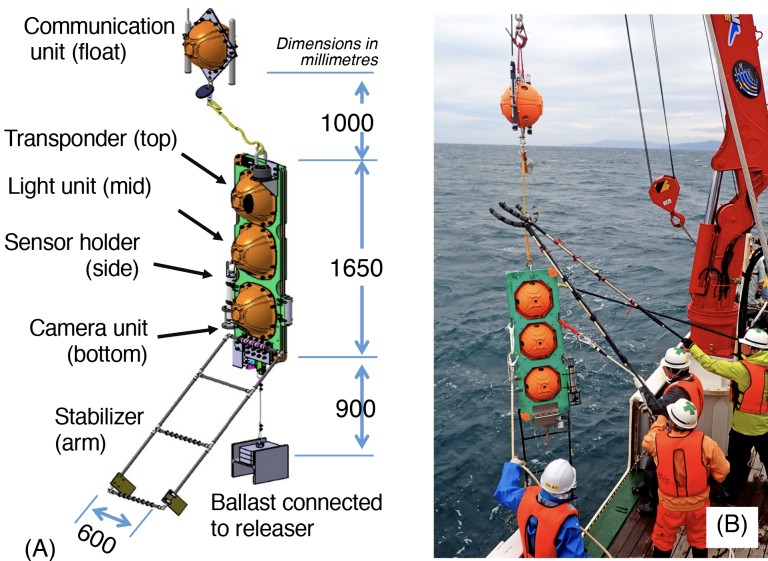

**Figure 2** **Edokko Mark 1.** (A) Composition and (B) being deployed.

seconds at the beginning when the lighting conditions were unstable just after power supply started. Care was taken during the image extraction to set the start and end times to include only the time period for which lights were on and the illumination was stable. A total of 49 images were extracted from each 1-min recording-interval movie (.mpeg) file. The brightness values (0–255) for each color channel (red, green, blue: RGB) were calculated for each and every pixel in the image and the average values for each channel were then combined using the following formula: L = B*0.114478+G*0.586611+R*0.298912, where L is luminance, according to the International Telecommunication Union standard ITU-R BT.601 (*International Telecommunication Union, 2011*). An average luminance value was then calculated for each image by averaging the values for all pixels, and the 49 values for the 49 images were then averaged to calculate the average luminance of each 49 second-long video file.

A vertical profile of seawater properties at the deployment location for the Edokko Mark 1 was made using a XCTD profiler (XCTD-4, Tsurumi-Seiki Co., Ltd.). For a comparison to the baseline, a CTD data profile, RF-6374 (34°39.12′N–137°00.71′E), was derived from open data of the 50-years 137°E hydrographic section maintained by the Japan Meteorological Agency (e.g., *Oka et al., 2018*; *Japan Meteorological Agency, 2019b*).

## RESULTS

The 40 hours of intermittent monitoring with Edokko Mark 1 successfully detected in situ deep-sea turbidity currents, probably associated with Typhoon Hagibis. Edokko Mark 1 monitored and recorded the temporal variation of pressure, turbidity, temperature, and salinity (Fig. 3), as well as recording video images (Figs. 4 and 5) (Supplementary Video File). Oscillation of water pressure at the seafloor corresponded to the tidal cycle, while the

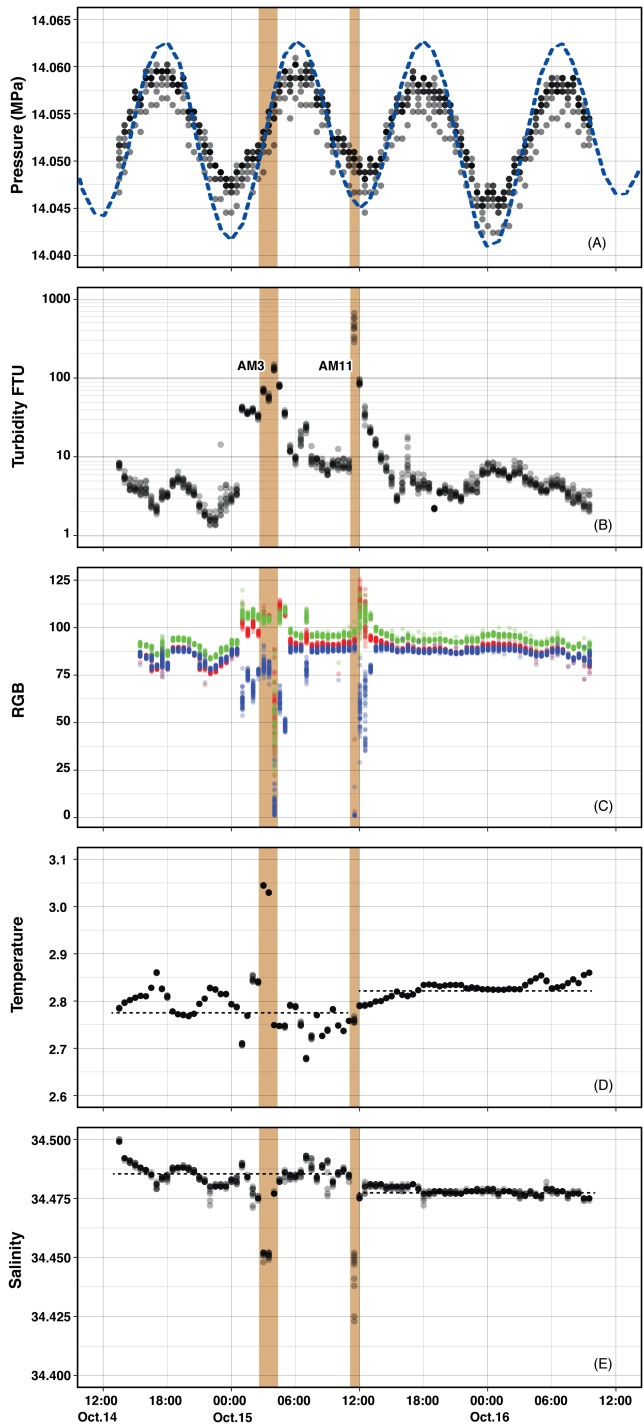

**Figure 3** **Temporal variation of parameters monitored by Edokko Mark 1.** (A) Pressure with surface tide level, (B) turbidity, (C) RGB values of video images, (D) temperature, and (E) salinity.

approximate mean pressure of 14.05 MPa represents the water depth of 1,370 m where the Edokko Mark 1 was deployed.

Seawater turbidity varied drastically during the observation period (Fig. 3). The turbidity stayed at a low level between 1-5 FTU with slight fluctuations until the end of October 14.

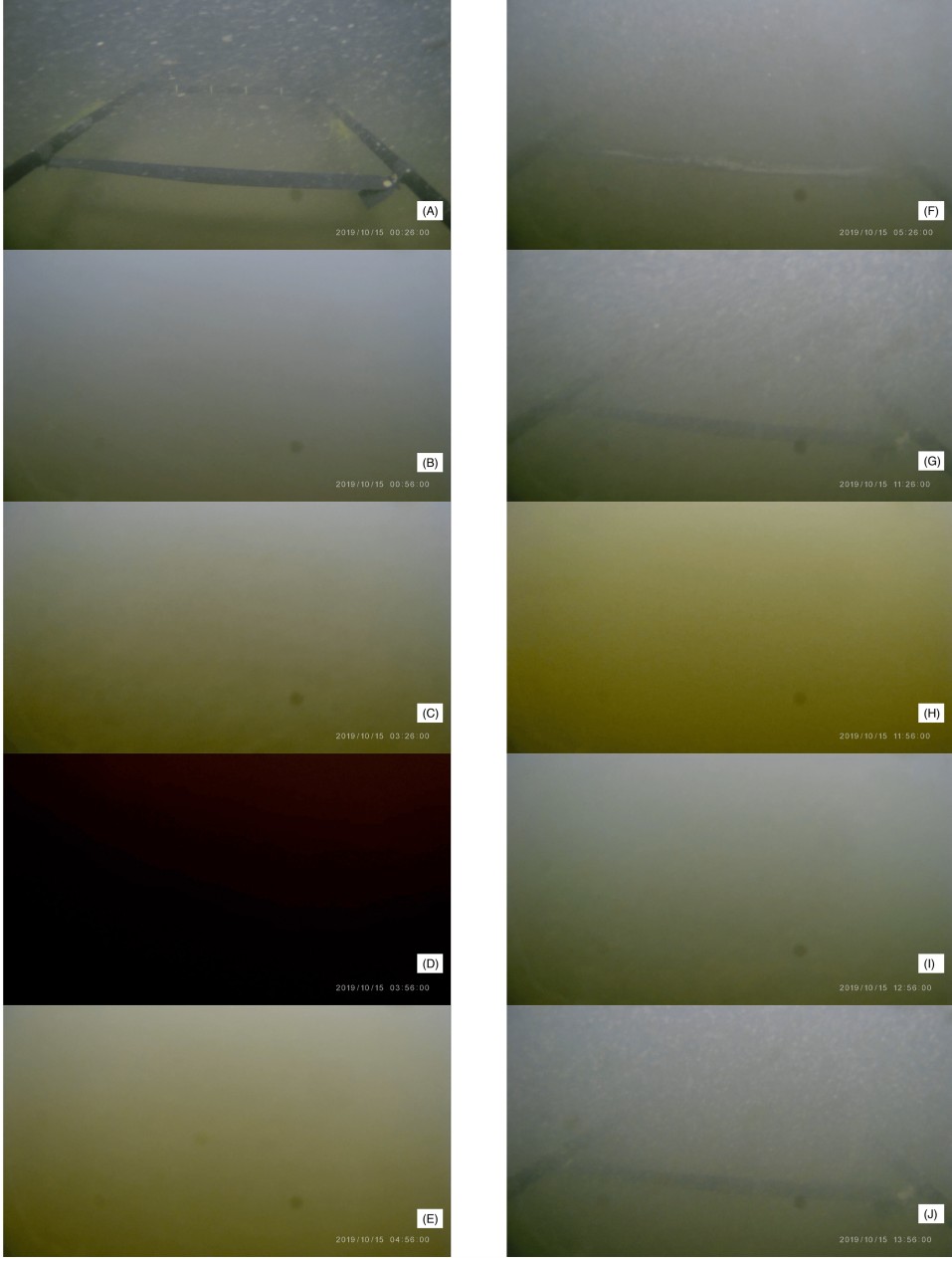

**Figure 4 Representative images captured by the video camera on Edokko Mark 1.** Time of extracted images are shown on the frame grabs (A–J). A black horizontal bar was placed ∼50 cm away from the video camera for distance indication.

From 00:55 to 14:55 of October 15, a high turbidity of >10 FTU was observed. Eventual spikes over 50 FTU were recorded at 02:55–04:26 and 11:26–11:56, hereafter referred to as the AM3 event and AM11 event, respectively. There was a relatively calmer period between 07:55–10:56. After 14:55 of 15 October, the turbidity again dropped to a low level until the observation ended on October 16. The timings of turbidity increment at 00:55 and the

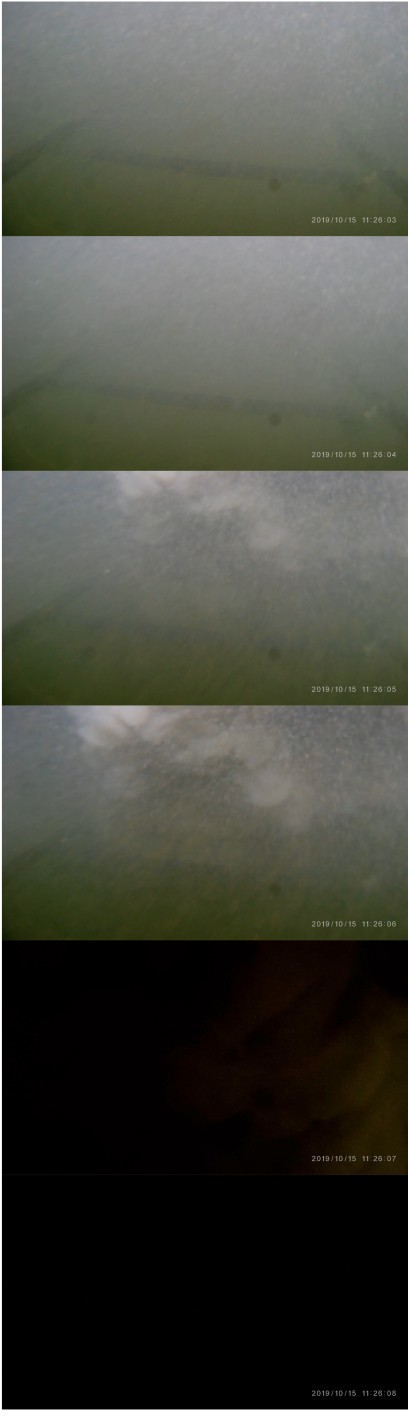

**Figure 5** **Sequential images during six seconds when a turbidity current struck Edokko Mark 1.**

AM3 event corresponded to those of the low tide, while the turbidity peak of the AM3 event was 3 hours after the last low tide.

Magnitude of seawater cloudiness in the video images recorded by the camera (Figs. 4 and 5) corresponded well to turbidity levels recorded by the turbidity sensor (Fig. 3). At 00:26 on 15 October, the horizontal bars of the Edokko Mark 1's arm, located 30 cm away from the front of the main body, can be identified (Fig. 4A). During the period from 00:56 on 15 October, where strong turbidity was detected, video images confirmed a storm of brownish particles and the bars became indiscernible (Figs. 4B, 4C). When the turbidity reached over 100 FTU at 03:56, the video camera blacked out (Figs. 4D, 4E), suggesting that the turbidity current struck the camera and light on Edokko Mark 1 after 03:26. Following a relatively calm period (Figs. 4F, 4G), another black-out situation occurred at 11:26 on 15 October during the AM11 event, when the video camera captured the moment where a turbidity current struck the Edokko Mark 1 (Figs. 4H, 5) (Supplementary Video File). After 13:55 on 15 October, when turbidity values were lower than 10 FTU, the arm was again identifiable in the video images (Figs. 4I, 4J, 4K). The RGB characteristics of the images demonstrated a decrease in blue component when the seawater turbidity increased (Fig. 3), indicating the predominance of brownish particles.

Both temperature and salinity, monitored by the Edokko Mark 1, were not stable during the observation period. The temporal variations of temperature and salinity generally yielded a mirror image of each other, even during the temperature peak at 02:55–03:26 on 15th October, in the AM3 event (Fig. 3) just before the extreme turbidity over 100 FTU was recorded at 03:55. At the AM11 event, however, only salinity decreased while temperature remained relatively stable. The degree of variation seen in both temperature and salinity were large leading up to the AM11 event, and became more stable after that. Quantitatively, the mean temperature (shown with standard deviation) before the AM11 event was 2.78 ± 0.04 °C and 2.83 ± 0.02 °C after it. The mean salinity, on the other hand, was 34.485 ± 0.005 before the AM11 event and 34.478 ± 0.002 after. Note that the temperature and salinity data during the AM3 and AM11 events were eliminated from the calculation. The decreases in salinity, measured by conductivity using the CTD, were observed during the high turbidity events. These are attributable not only to seawater with lower salinity, but also to increased concentration of suspended particles due to suspended particles being less conductive than seawater (e.g., *Wang et al., 2020*). On the other hand, the thermometer is inert to the changes in concentrations of suspended particles and was thus unaffected.

The relationships between the timing of low tide, increase in turbidity and temperature, and video capture of destructive turbidity current differed between AM3 and AM11 events. Before the AM3 event, the lowest tide occurred at midnight and was followed by increase in turbidity at around 01:00 and temperature at around 02:00 (Fig. 3), both reaching highest values at the AM3 event. Before the AM11 event which happened at the same time as a low tide, however, no such signs were detected in the sensors prior to the visual confirmation of the turbidity current, following which the turbidity declined consistently.

The characteristics of potential temperature and salinity observed by the Edokko Mark 1 during the turbidity current observation generally agreed with those of the ambient water column at the Edokko Mark 1's deployment locality, as observed by the XCTD, as well as the reliable reference data RF-6374 (Fig. 6). Slight offsets of potential temperature and salinity

characteristics among data from the Edokko Mark 1, XCTD, and RF-6374 are probably due to the insufficient calibration of one or more devices. Consistencies in the trends of potential temperature and salinity variation between the seafloor-deployed Edokko Mark 1 and vertical profilers strongly suggests the displacement of the generally-stratified deep seawater around the Edokko Mark 1. Outside the two events, the potential temperature and salinity values recorded by the Edokko Mark 1 showed slight fluctuation and corresponded to those of the ambient deep-seawater at 1,300–1,380 m depth observed by the XCTD, close to the seafloor depth of 1,370 m (Fig. 6). Potential temperature and salinity recorded by the Edokko Mark 1 during the AM3 event also followed the trend of the ambient seawater and corresponded to those of seawater approximately 200 m shallower than Edokko Mark 1′s location (Figs. 6A–6D). Although the properties of the seawater during the AM3 event appears to be attributable to 1 % contribution of surface seawater with a salinity of 31 and temperature of 25 °C into the bottom water (Fig. 6E), the direct bimodal mixing between surface and bottom waters is unlikely due to some mixing with seawater at intermediate depths during the downslope transportation being inevitable. On the other hand, the potential temperature and salinity during the AM11 event are too far deviated from those of the water column, suggesting false conductivity signals resulting from the high concentration of suspended particles, as discussed above.

## DISCUSSION

Turbidity currents can be considered to be the driving force for the vertical stirring of deep-sea water observed in the CTD data during this study, since, in general, low-density shallow seawater cannot sink down to great depths without an external driving force. In support of this, previous observations by cabled observatories have consistently recorded increases of seawater temperature at the seafloor accompanied with turbidity currents (*Mikada et al., 2006*; *Kasaya et al., 2009*; *Zhang et al., 2018*). Since a turbidity current is a composite of sedimentary particles suspended in a seawater matrix, the density is naturally higher than the density of interstitial water alone and causes the shallow seawater to sink to great depths (*Kao et al., 2010*). This mechanism is the most plausible explanation for the current CTD observations by the Edokko Mark 1. As interstitial water density of the typhoon-induced turbidity current during the AM3 event was approximately 0.05 kg/m$^3$ lower than that during periods outside of the two events (Fig. 6D), the turbidity current was able to obtain sufficient density from the suspended particles. The suspended particles of > 50 FTU, corresponding to 0.05 kg/m$^3$ under the model definition (see 'Materials & Methods'), can produce sufficient gravity in the whole turbidity current, resulting in the sinking of seawater into the depths of Suruga Bay from at least 200 m above the seafloor. We would, however, caution that the 1:1 model conversion of FTU to g/m$^3$ is unlikely to be accurate for natural seawater due to variable densities of suspended particles.

The deep-sea turbidity current observed in this study was probably induced by the record-breaking Super Typhoon Hagibis (2019). Although the deep-sea turbidity current observed here occurred two days after Hagibis passed over Suruga Bay, it has previously been reported that deep-sea turbidity current occurred even several days after a cyclone passes (*Pope et al., 2017*). Although earthquake-induced deep-sea disturbances have been

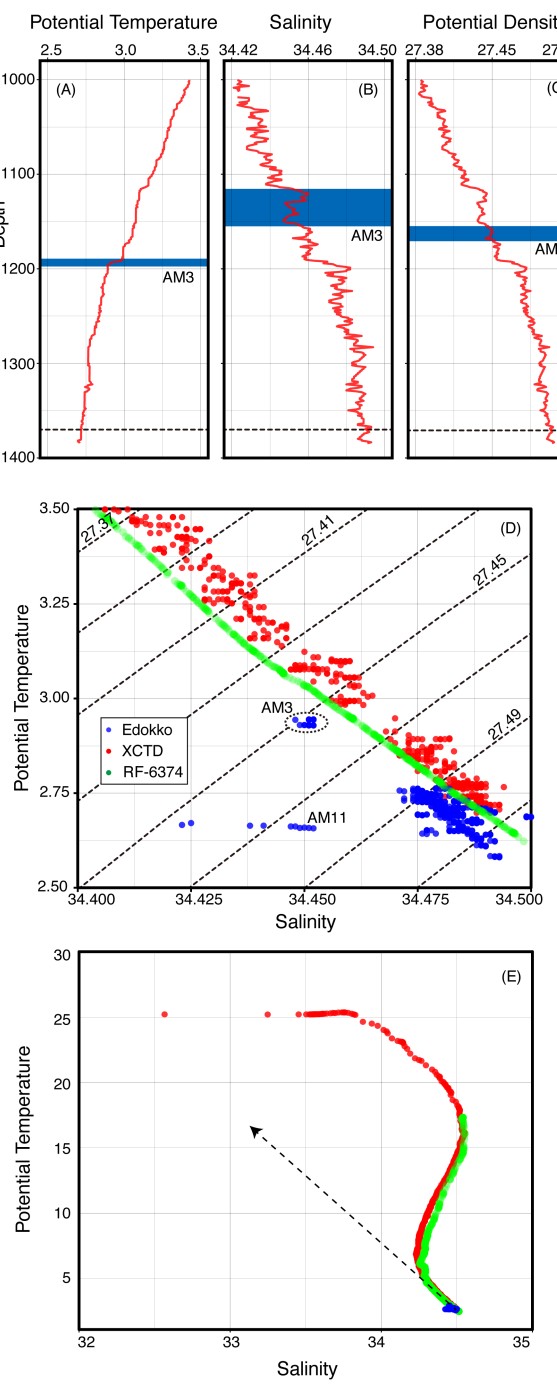

**Figure 6** Seawater characteristics observed by XCTD (A–C), Edokko Mark 1, and the reference RF-6374 (D–E).

observed to date (*Mikada et al., 2006*; *Kasaya et al., 2009*; *Kawagucci et al., 2012*; *Noguchi et al., 2012*), no significant earthquake (M>1) occurred around the Suruga Bay area during Edokko Mark 1's monitoring period. Even though an M5 earthquake occurred at 200 km
east of Suruga Bay at 18:00 on 12 October [JMA 2019], this is both too early and too far away to have been the cause for the turbidity current observed in Suruga Bay by the Edokko Mark 1.

Pin-pointing the trigger mechanism for the deep-sea turbidity current observed herein is challenging, due to the limited availability of information. The turbidity current was observed from a single monitoring location and a single vertical point of the deployed system that could not measure the direction and speed of the in situ current, with no supplemental observation at the river and coast to confirm the exact time and location of the turbidity current release. A known major trigger mechanism for the turbidity current associated with typhoon is river flood (*Liu et al., 2012*; *Lintern, Hill & Stacey, 2016*; *Clare et al., 2016*; *Hage et al., 2019*). The torrential rainfall from Hagibis on October 12 (*Takemi & Unuma, 2020*) indeed resulted in intensive flooding of rivers in the Izu Peninsula, subsequently flowing into the eastern part of Suruga Bay (*Shizuoka Local Meteological Office, 2019*). However, the water level of these rivers returned to normal before the noon of October 13 (*Shizuoka Local Meteological Office, 2019*), over 24 hours before the turbidity flow event we observed on October 15. This long lag suggests that the flooding associated with Hagibis itself could not trigger the observed turbidity current. On the other hand, turbidity in the river water may have remained high until October 15 after flooding has ceased, and this hyperpycnal river flow could have continued to supply sufficient suspended particles to trigger the turbidity current. If this combined with low tide at midnight of October 15 to trigger the turbidity current seen during the AM3 event, it would have had a horizontal velocity of 1 m/s, estimated from the duration of 3 hours and the distance of approximately 11 km between the Heda river mouth and the location of our Edokko deployment (Fig. 1A). This is comparable with the typical velocity seen in turbidity currents (e.g., *Khripounoff et al., 2012*). The seawater property demonstrating entrainment with seawater from 200 m shallower is also not inconsistent with this trigger mechanism when located at a coastal region. Nevertheless, no concrete evidence is available to verify this scenario.

Another possible trigger for the turbidity current observed is submarine slope failure. It has previously been reported that the rapid accumulation of seafloor sediment associated with tropical cyclones can eventually cause submarine slope failure, resulting in the runoff of a deep-sea turbidity current, even several days after a cyclone passes (*Pope et al., 2017*). The torrent of flooding, muddy streams in the forest-rich Izu Peninsula up until the noon of October 13 delivered an unusually large amount of terrestrial soil into the seafloor of Suruga Bay, and a part of this accumulation may be further transported into the deep sea before the start of our observation on October 14. Such unconsolidated sediments could serve as precursors of the turbidity current observed on October 15. Particularly, the turbidity current at the AM11 event which occurred synchronously at low tide without changes in seawater properties would have occurred locally, possibly triggered by a local slope failure around the Edokko Mark 1, rather than being directly triggered by turbid waters from the river.

## CONCLUSIONS

We successfully recorded a deep-sea turbidity current probably induced by the Super Typhoon Hagibis and vertical stirring of deep-sea water due to this turbidity current, based on observations by the easy-to-use, autonomous monitoring system Edokko Mark 1. These spontaneous events are difficult to observe without in situ long-term monitoring, and this capacity will help illuminate the diverse forms and mechanisms of turbidity currents in the deep ocean.

## ACKNOWLEDGEMENTS

The authors are grateful to Drs. Chong CHEN, Kazuya KITADA, Satoshi OSAFUNE, and Shiro MATSUGAURA for supporting cruise operation, data processing, and manuscript editing. This work was conducted as a part of Cross-ministerial Strategic Innovation Promotion Program (SIP) "Innovative Technology for Exploration of Deep-Sea Resources" (Lead agency: JAMSTEC). The authors are deeply appreciative of the cooperation of the Pacific Islands members who joined the capacity building cruise of SIP.

## ADDITIONAL INFORMATION AND DECLARATION

### Funding

This work was supported by the Japanese Council for Science, Technology, and Innovation (CSTI), Cross-ministerial Strategic Innovation Promotion Program (SIP) "Innovative Technology for Exploration of Deep-Sea Resources" (Lead agency: JAMSTEC). The funders had no role in study design, data collection and analysis, decision to publish, or preparation of the manuscript.

### Grant Disclosures

The following grant information was disclosed by the authors:
Japanese Council for Science, Technology, and Innovation (CSTI).
Cross-ministerial Strategic Innovation Promotion Program (SIP) "Innovative Technology for Exploration of Deep-Sea Resources" (Lead agency: JAMSTEC).

### Competing Interests

Ayu Takahashi and Sangkyun Lee are employed by JGI, Inc. and Ken-ichiro Nishibayashi, Hiroyuki Yokooka and Shotaro Nishi are employed by IDEA Consultants, Inc.

### Author Contributions

- Shinsuke Kawagucci, Dhugal J. Lindsay and Hiroyuki Yamamoto conceived and designed the experiments, analyzed the data, prepared figures and/or tables, authored or reviewed drafts of the paper, and approved the final draft.
- Tetsuya Miwa and Ayu Takahashi conceived and designed the experiments, performed the experiments, authored or reviewed drafts of the paper, and approved the final draft.
- Eri Ogura analyzed the data, prepared figures and/or tables, authored or reviewed drafts of the paper, and approved the final draft.

- Kenichiro Nishibayashi, Hiroyuki Yokooka and Shotaro Nishi performed the experiments, authored or reviewed drafts of the paper, and approved the final draft.
- Sangkyun Lee conceived and designed the experiments, performed the experiments, prepared figures and/or tables, authored or reviewed drafts of the paper, and approved the final draft.

## Data Availability

The raw data are available in the Supplemental Files.

## Supplemental Information

Supplemental information for this article can be found online at http://dx.doi.org/10.7717/peerj.10429#supplemental-information.

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
