# Peer review of "Deep-sea water displacement from a turbidity current induced by the Super Typhoon Hagibis"

_PeerJ, doi:10.7717/peerj.10429_

## Round 0.1 · original submission · Major Revisions

Please provide a point-by-point response to the reviewers' comments and indicate which changes you make to address them.

·

Basic reporting

Writing is clear and professional throughout. Literature reference could be further extended (see review attached).

Experimental design

Method is rigorous with sufficient detail. Aims are clear.

Validity of the findings

Data seems sound. Conclusion well stated, but could be improved (see review attached).

Additional comments

see review attached

·

Basic reporting

The manuscript discusses direct observations of turbidity current events most likely induced by Super Typhoon Hagibis, in Suruga Bay, Japan.

The manuscript is well written in general and provides adequate literature references and context. I would recommend avoiding idioms such as for instance "wreaking havoc" in the abstract. I also recommend that the authors check the accepted date format in PeerJ. For instance, "14 October" should most likely be "October 14" instead.

Figures are clear and add value to the manuscript. A schematic of the different sensors mounted on Edokko Mark 1 could be added to figure 2, as it is not essential to the discussion in its present form. The caption for figure 6 should mention if this is for the AM3 or AM11 event.

Experimental design

Direct, in-situ measurements of turbidity current are always relevant and interesting, and the methods described in the paper are, to the best of my understanding, technically adequate. The turbidity current is observed in a single location, which makes it impossible to estimate its kinematic properties and to infer its propagation speed. This is obviously not something that can be addressed in the present manuscript, but I would be interested in the future to see the methods described here applied at different locations downslope from the possible release point of the turbidity event.

Validity of the findings

The paper clearly demonstrates the existence of the turbidity events, well supported by direct measurements. However, the authors additionally state that "The
temperature and salinity characteristics during these high turbidity events matched
independent measurements for shallow water layers in Suruga Bay at the same time,
strongly indicating that the turbidity current caused mixture of the bay’s water column". I find this statement confusing, as it suggests that the turbidity current generates vertical mixing throughout the water column, at the location where it is measured, which I do not believe to be the case. If my understanding is correct, the turbidity current transports warmer interstitial fluid from a shallow region of the Suruga Bay to the deeper region where Edokko Mark 1 is located, downslope of the canyon. This is made possible, as the authors point out, by the fact that the suspended particles are presented in high enough concentration for the mixture to have a higher density than the deep ambient fluid. Downslope transport of interstitial fluid by turbidity currents is completely expected and is, in my understanding of the term, significantly different from predominantly vertical mixing, which would refer to convection of warmer fluid directly above Edokko Mark 1 down to the seabed, which I do not believe is the case. I suggest that the authors revisit the statement quoted at the top of this paragraph and clarify what is meant by "vertical mixing".

Because of uncertainty with the salinity measurement during the turbidity event, it seems challenging for the authors to provide a more quantitative assessment of the density of the interstitial fluid, as well as overall mixture density, of the turbidity current. Because the time and location of release of the turbidity current is not known, it is also impossible to estimate the speed of the current front and thus to estimate its properties based on simple models. Because this quantitative analysis cannot be performed, I strongly encourage the authors to improve their discussion of mixing and transport, as discussed above.

Additional comments

My understanding of the results presented is that the turbidity current does not generate a large scale vertical mixing of the water column in the Suruga Bay, but instead, as is expected of a turbidity current, transports interstitial fluid from the shallower release site to deeper parts of the bay, following the slope of the canyon topography. If so, the discussion should be modified accordingly. If there is indeed large scale mixing throughout the water column, the discussion should be improved to better demonstrate this statement, and show how the results prove this is the case.

Minor comment: The last paragraph of the discussion does not refer to the results presented in the results section, and reads more like an introduction. I recommend that the authors move this paragraph to the introduction, and develop the points identified above in my comments in the discussion instead.

Reviewer 3 ·

Basic reporting

no comment

Experimental design

no comment

Validity of the findings

no comment

Additional comments

Kawagucci et al. present an interesting work studying a direct observation of the deep-sea turbidity current triggered by the Super Typhoon Hagibis (2019). The manuscript is thus worthy of such rapid publication but should be done after a few necessary revision works below:

Reads unclear whether the authors suggest canyon helps the turbidity mixing observed by, for example, "canyon flushing event." The authors should convey the mechanism on how Hagibis triggers and generates the turbidity current the authors observed. Several papers studying Japan and NZ below are helpful:
Kioka, A., Schwestermann, T., Moernaut, J. et al. Megathrust earthquake drives drastic organic carbon supply to the hadal trench. Sci Rep 9, 1553 (2019). https://doi.org/10.1038/s41598-019-38834-x
Mountjoy, J.J., Howarth, J.D., Orpin, A.R., et al. Earthquakes drive large-scale submarine canyon development and sediment supply to deep-ocean basins. Sci Adv 4, eaar3748 (2018). https://doi.org/10.1126/sciadv.aar3748

Specific comments

L42: "This" -> "A turbidity current"
L47: "matched" -> "agreed with"
L48: "strongly indicating" -> "indicating"
L52: "greatly contribute to" -> "contribute to"
L64-67: There is relevant literature studying direct observation of deep-sea turbidity current:
Paull, C.K., Talling, P.J., Maier, K.L., et al. Powerful turbidity currents driven by dense basal layers. Nat Commun 9, 4114 (2018). https://doi.org/10.1038/s41467-018-06254-6
L85: Yes I remember the cancellation was a great shock ...
L88: Should specify the time zone here. "18:00" should be "18:00 (UTC+9:00)" given the authors may use Japan standard time. Modify Figure 1 accordingly
L92-96: Long sentence
L119: Address the location more specifically (e.g., the mouth of Heda Canyon)
L123: I kannt find the Heda River in Figure 1. Could you please show it in Figure 1?
L164: "14 October" should be "16 October?"
L187-190: Hard to believe the temperature value is such high precise. Perhaps to better to round up at least to the second decimal place...
L262-265: This should be moved to the Discussion session because it should not be a conclusive message of the study.

---

## Round 0.2 · Minor Revisions

Please address the comments in the annotated manuscript of reviewer #1.

·

Basic reporting

Clear and unambiguous, although I have added some minor textual suggestions in the pdf attached.

Experimental design

original primary research within aims and scope of the journal.

Validity of the findings

Conclusions are well stated, linked to the original research question and limited to supporting results.

Additional comments

I really enjoyed reading the manuscript again. I have provided some small suggestions in the sticky notes on the attached pdf, but I do not see any issue that would prevent me to recommend publication of the manuscript. Looking forward to see more results coming from this new mooring!

All the best,
Matthieu

·

Basic reporting

My comments have been addressed, nothing to add.

Experimental design

My comments have been addressed, nothing to add.

Validity of the findings

My comments have been addressed, nothing to add.

Additional comments

The authors have addressed my comments and concerns throughout the manuscript, which has been greatly improved.

---

## Round 0.3 · Minor Revisions

Many thanks for your revised manuscript. This looks all fine. I apologise for having overlooked some necessary technical corrections. Could you please also implement the following corrections:

1) Define the meaning of the abbreviation FTU (formazine turbidity unit).

2) On l. 292, change "sec" to the SI unit symbol "s".

3) Remove the 10 occurrences of a "tilde" sign before a quantity, apparently used to indicate approximations. Please note that all measured values should be rounded appropriately and reported according to their precision. An approximation symbol is then not needed. Approximation symbols really make sense only for mathematically exact values, e.g. √2 ≈ 1.4, or e ≈ 2.71, but no such values are referred to in the manuscript. In any case, the correct approximation symbol would be "≈" (two wavy lines).

---

## Round 0.4 · accepted · Accept

I am very happy to accept your manuscript for publication.